**www.cambridge.org/ext**

Diversity loss; biocultural diversity; Indigenous languages; language revitalization; macroecology; hotspots

**Author for correspondence:**
Lindell Bromham,
Email: Lindell.Bromham@anu.edu.au

# Language endangerment: Using analytical methods from conservation biology to illuminate loss of linguistic diversity

Lindell Bromham 

Macroevolution and Macroecology Group, Research School of Biology, Australian National University, Canberra, ACT 0200, Australia

## Abstract

Language diversity is under threat, with between a third to a half of all languages considered endangered, and predicted rates of loss equivalent to one language per month for the rest of the century. Rather than reviewing the extensive body of linguistic research on endangered languages, this review focuses specifically on the interdisciplinary transfer of methods developed in conservation biology, macroecology and macroevolution to the study of language endangerment and loss. While the causes of language endangerment and loss are different to those for species, studying patterns of diversity of species and languages involves similar analytical challenges, associated with testing hypotheses and identifying causal relationships. Solutions developed in biology can be adapted to illuminate patterns in language endangerment, such as statistical methods that explicitly model phylogenetic nonindependence, spatial autocorrelation and covariation between variables, which may otherwise derail the search for meaningful predictors of language endangerment. However, other tools from conservation biology may be much less use in understanding or predicting language endangerment, such as metrics based on International Union for Conservation of Nature (IUCN) criteria, population viability analysis or niche modelling. This review highlights both the similarities and the differences in approaches to understanding the concurrent crises in loss of both linguistic diversity and biodiversity.

## Impact statement

The year 2022 marks the beginning of the UNESCO Decade of Indigenous Languages, declared to draw attention to the catastrophic loss of global linguistic diversity, and the need to act swiftly to safeguard minority languages. To what extent can analytical tools developed in conservation biology to document patterns and causes of species diversity be applied to understanding language endangerment and loss? Many methods from biology can be adapted with due recognition of both the similarities and the differences between language loss and species endangerment. While global-scale analyses cannot capture key local processes contributing to language endangerment and loss, they can reveal some general patterns that may help to identify general contributors to loss of linguistic diversity.

## Languages and species: Similarities and differences

This review concerns the application of methods developed in conservation biology, macroecology and macroevolution to the patterns and causes of language endangerment. This is not a general review of language endangerment because there are many excellent reviews written by experts in linguistics (e.g., Maffi, 2002; Romaine, 2007; Rehg and Campbell, 2018; Evans, 2022). Instead, it is focused specifically on the way that methods originally developed in biology have been adapted and applied to endangered languages.

Since the beginnings of the disciplines of both evolutionary biology and historical linguistics, it has been recognised that the processes and patterns of species diversity and language diversity are 'curiously the same' (Darwin, 1871). Examples from language evolution were used as a convincing demonstration that Darwin's proposed mechanism of evolution could generate diversity over time (e.g., Lyell, 1863; Schleicher, 1863): variations arise in individuals, which can be copied and may rise in frequency in a population, eventually replacing the previous variant, contributing to the accumulation of changes that ultimately generates separate mutually unintelligible languages or non-interbreeding species (see Bromham, 2017). The study of both language change and biological evolution is, by and large, built on this uniformitarian principle that change at the population level (microevolution) drives patterns of diversity (macroevolution). While there is a striking resemblance between language change and species evolution, there are also differences. For example, while genetic mutation is generally considered to be random with respect to fitness,

language change may be either random or directed: language change is driven by individuals who can make deliberate decisions, such as invention of a new word to suit a particular purpose, and can agree on population level change, for example, intentional regularisation of spelling or grammar. While genes are predominantly passed vertically from parent to offspring, language variants are acquired both vertically and horizontally, from other members of the population. In some cases, the differences in mechanisms of change can be incorporated within existing biological models; in other cases, methods from biology may require modification to fit language change.

There are also both similarities and differences in the current catastrophic loss of both biodiversity and linguistic diversity (Sutherland, 2003). The International Union for Conservation of Nature (IUCN) provides a global reference for threatened species known as the Red List. Each listed species has been evaluated by experts, with threat status largely based on population size and area, along with patterns of decline, reduction in area and fragmentation of populations. Overall, 28% of assessed species are considered threatened (i.e. falling into IUCN categories Vulnerable, Endangered and Critically Endangered), but some groups have relatively more threatened species, for example, 41% of amphibians, 37% of sharks and rays and 33% of reef corals (IUCN, 2022). Estimates of the percentage of languages that are endangered vary between 43% and 63% (see Table 1). One important difference between endangered species and languages is in the amount of undescribed diversity. At least 90% of all spoken languages have been formally evaluated and given an endangerment score, although many signed languages remain poorly documented or unassessed. In contrast, 142,500 species have been assessed by the IUCN, less than 10% of all described species, and less than 2% of the estimated number of species on earth (Mora et al., 2011; Larsen et al., 2017).

One notable difference between the study of endangered languages and species is that biologists do not have to consider how endangered species feel, but researchers who work on endangered languages must be sensitive to the role that language can play in personal identity and cultural recognition (Auger, 2016), and be mindful of their choice of language in describing language loss, recognising the range of strong emotions that can be associated with discussions of endangered languages. While languages with few or no speakers are sometimes referred to as 'moribund', 'dormant' or 'extinct' (Table 1), some community members find these terms confronting and distressing. Consultation on appropriate, respectful terminology is essential. The growing number of language revitalisation programmes demonstrates that language 'extinction' does not necessarily mean permanent silence (Hinton, 2003; Grenoble and Whaley, 2005; Hinton et al., 2018; O'Grady, 2018), leading to a preference for terms such as 'Sleeping' to describe languages that currently have no fluent speakers (Leonard, 2008; Zuckermann and Walsh, 2011). There is a growing emphasis on Indigenous perspectives on the study of endangered languages, and increasing recognition that academic discussion about endangered languages should include the voice of speakers of endangered languages (McCarty, 2003; Davis, 2017; Meakins et al., 2018).

Like biodiversity, language diversification is a continuous process of generation of new languages, sometimes accompanied by the loss or replacement of other languages. Languages, like species, can be lost if there are no new generations using the language to communicate (e.g., Etruscan, a widespread language of Italy which ceased to be used around 2,000 years ago, and left no descendant languages). Like species, languages can also undergo so much change over time that the ancestral language is sufficiently different to be considered separate language from its descendants (e.g., Latin, a language of a similar time and place as Etruscan, is no longer spoken but gave rise to many modern languages including Italian). Furthermore, like biodiversity, while language turnover is a natural part of evolution, the pace of loss has accelerated in recent times. Catastrophic loss of language diversity has occurred through colonisation, conflict, cultural suppression, socioeconomic change and migration, and the loss is ongoing, with many languages now only spoken by a dwindling number of elders. Language loss, like species loss, robs us of richness, variety and beauty. Each language represents a unique human invention, a creative expression of a rich culture and a storehouse of knowledge about the history, culture and environment of its speakers.

Language loss, like species loss, can occur when the speaker population is catastrophically reduced, for example, through violence or disease (Bowern, 2022). However, in contrast to biodiversity, language loss also occurs through language shift, when speakers cease to speak their heritage language, or their children no longer learn it, as they adopt a different language to communicate. Decline through language shift makes language loss much more complicated than species extinction. Language shift may be forced upon a population (e.g., removal of children to prevent language transmission or punishment for speaking an Indigenous language), or it may occur through lack of support for a minority language (e.g., use of an alternative dominant language in schooling, commerce and employment), or it may accompany economic, social or geographic movement (e.g., migration or colonisation). Language shift complicates the relationship between population size and number of speakers, and this makes some techniques from conservation biology difficult to apply to language endangerment.

## Population size and endangerment

Unlike species, where population counts are based on number of mature individuals, population size for languages is generally based on the number of fluent first-language (L1 or 'mother tongue') speakers. Number of L1 speakers might not reflect the number of fluent speakers of a language. Some languages have more second-language (L2) speakers, for example, Urdu has more than twice as many L2 speakers as L1. Some languages have only L2 speakers, such as Fanagalo (or Fanakalo), a 'lingua franca' from Africa which aids communication between people with different L1 languages, particularly in the mining industry (Ravyse, 2018). Fanagalo is primarily used in the workplace or as a 'market language', not learned by children as their first language.

Speaker population size has been used as an indicator of language endangerment. For example, Amano et al. (2014) used IUCN criteria to identify endangered languages based on small speaker populations (<1,000), restricted range (<20 km$^2$) and speaker population declines (see also Sutherland, 2003). However, there are several limitations with using range and population size as a primary indicator for language endangerment. Quantitative data on declines in speaker number are available for relatively few languages, and for this reason, Amano et al. (2014) could include only 9% of the world's languages in their analysis. More generally, population size is not always a good indicator of language vitality.

The number of L1 speakers cannot, by itself, be used as a metric of endangerment. A language with a very small L1 population size can be stable if it is being consistently learned by each new

**Table 1.** Databases and scales for language endangerment with the estimated percentage of endangered languages. Note that the while the words 'extinct', 'dead', 'moribund' and 'dormant' are used in these scales, they are considered inappropriate by many communities and workers in the field of endangered languages, who prefer alternative terms such as 'Sleeping', on the grounds that even a language with no current L1 speakers may be revitalised. The Expanded Graded Intergenerational Disrupted Scale (Lewis et al., 2013) is based on the number of L1 speakers, domains of use (e.g., government, trade, education and home), intergenerational transmission (e.g., whether being actively learned by children), official recognition and stability (whether the language is stable or declining; Grenoble and Whaley, 1998; Lewis and Simons, 2010). UNESCO ranks languages into six levels based on speaker population size, intergenerational language transmission, proportion of speakers within the total population, community attitudes, shifts in domains of use, educational materials and documentation, institutional recognition and government policies (Moseley, 2010). The Catalogue of Endangered Languages (ELCat; University of Hawaii at Manoa, 2019) uses the Language Endangerment Index (LEI), based on intergenerational transmission, domains of use, number of speakers and whether the population is increasing or decreasing (Lee and Van Way, 2016), along with an uncertainty score based on the reliability of information available (Lee and Van Way, 2018). The Agglomerated Endangered Scale (AES; Hammarström et al., 2019) takes information from other scales, preferencing LEI first, then UNESCO, then EGIDS (Hammarström et al., 2018). Alignment between scales is based on Hammarström et al. (2018)

| Scale | EGIDS | | | AES | UNESCO | LEI |
|---|---|---|---|---|---|---|
| Used by | Ethnologue | | | Glottolog | Atlas of the World's Languages in Danger | ELCat |
| Accessibility | Paid licence | | | Open access | Open access | Open access |
| Languages | 7,151 | | | 8,565 | 2,500 | 3,459 |
| % Endangered | 43% | | | 63% | 50% | 43% |
| Scale | 0 | International | Widely used in trade, knowledge exchange and international policy | Not endangered | Safe | Safe |
| | 1 | National | Used in education, work, mass media and national government | | | |
| | 2 | Provincial | Used in education, work, mass media and regional government | | | |
| | 3 | Wider communication | Used in work, mass media, but without official status | | | |
| | 4 | Educational | Vigorous use with standardisation and literature, and supported education | | | |
| | 5 | Developing | Vigorous use with literature but not widespread or sustainable | | | |
| | | Dispersed | Used in home country, standardised form and literature, but not promoted in education | | | |
| | 6a | Vigorous | Sustainably used for face-to-face communication by all generations | | | |
| | 6b | Threatened | Used for face-to-face communication within all generations, but losing users | Threatened | Vulnerable | Vulnerable |
| | 7 | Shifting | Child-bearing generation use among themselves, not transmitting to children | Shifting | Definitely endangered | Threatened |
| | | | | | | Endangered |
| | 8a | Moribund | Used only by grandparent generation and older | Moribund | Severely endangered | Severely endangered |
| | 8b | Nearly extinct | Limited use by grandparent generation only | Nearly extinct | Critically endangered | Critically endangered |
| | 9 | Dormant | Serves as a reminder of heritage identity, but use is symbolic | Extinct | Extinct | Dormant |
| | | Reawakening | Community working to establish users of once-dormant language | | | Awakening |
| | | | | | | Second language only |
| | 10 | Extinct | Not used and no current association with ethnic identity | | | Extinct |

generation. For example, Neko is spoken in a single village in Madang province, Papua New Guinea, with an L1 speaker population of around 650, yet it is not considered endangered due to stable transmission to younger generations. Languages with large numbers of L1 speakers can be endangered if they are not being actively learned by children. For example, Domari, a language

distributed across the Middle East and North Africa, is considered severely endangered despite having with over a quarter of a million L1 speakers, because it is predominantly spoken by elderly people. If a language is spoken only by adults and not learned by children, then the number of L1 speakers must decline, and eventually go to zero when the current L1 speakers have died. However, language revitalisation can lead to increase in the number of L2 speakers, and, potentially, a new generation of child learners (Hinton et al., 2018). For example, the Squamesh nation in Canada has more than 4,000 community members but there are few remaining L1 speakers of the language Sḵwx̱wú7mesh Sníchim; however, 10% of the community are actively learning the language through school and adult education programmes (Dunlop et al., 2018). If endangerment status is based only on L1 speakers, then languages with few or no child learners will be considered highly endangered, even if there are large numbers of fluent second-language speakers and adult learners. Fanagalo is learned by adults and has no L1 speakers, so it is rated as 'second language only', which is equivalent to 'extinct' in the Agglomerated Endangered Scale (Table 1), so Fanagalo is listed as 'extinct' in the global language catalogue Glottolog despite being actively used in a number of countries.

Quantifying the number speakers of any language is not straightforward (Moore et al., 2010) because languages do not necessarily have a one-to-one association with individuals. The majority of people in the world have more than one language, and counting the number of speakers of a language can be made difficult by subjective decisions about what constitutes fluency or how to define a 'native speaker'. In communities undergoing rapid language shift, there may be a generation of multilingual speakers who are familiar with their heritage language, and may have some degree of comprehension and usage but without the comfortable fluency of older generations (sometimes referred to as 'semi-speakers'; Dorian, 1977). There may also be people who were previously fluent in the endangered language, but may have become 'rusty' due to lack of opportunity to continue to use the language (sometimes referred to as 'rememberers' or 'latent speakers'). Yet, people who lack fluency may play a critical role in language vitality and revitalisation, and L2 speakers can contribute to language documentation and maintenance (e.g., Evans, 2001; Grinevald, 2003; Sallabank, 2018).

The counting number of L1 speakers is made challenging by the dynamic nature of language change, which can make the mapping of individuals to languages unclear. This is particularly the case when language change occurs rapidly in contact situations, resulting in new languages that blend aspects of several source languages. For example, several Indigenous Australian languages have distinct varieties spoken by younger generations. These new languages, such as Gurindji Kriol and Young People's Dyirbal, blend aspects of the traditional language of their community with elements from other languages along with innovations that are not found in either source language (Schmidt, 1985; Meakins et al., 2019). This situation can be viewed both as the loss or modification of the heritage language and the creation of a new language.

## Range size and language endangerment

Translation of IUCN-style endangerment metrics to languages is made problematic by differences in the way both population size and range size are recorded. Language range maps are not like species range maps. Species maps span the area where the species

has been reported to occur, which will overlap with many other species. However, language maps tend to be drawn as exclusive areas, so that there is typically only one autochthonous language at any point on the map, even if the population in that area includes L1 speakers of a diversity of languages, or if most of the community is multilingual. While it would be technically possible for language range databases to include all of the languages that occur in any given grid cell, in practice there are many obstacles in the way. One is the complexity of language distributions, which are shaped more by history and interaction than by environmental features. For languages associated with populations that live in settlements, such as towns and villages, language range maps in the form of polygons may be something of an abstraction. Growing recognition of the role of language maps in Indigenous land rights claims has highlighted that language ranges are complex and variable (e.g., Thom, 2009).

Another barrier to applying tools from ecology to language distribution data is that, unlike species distribution data, language range maps are currently not open access. There is only one comprehensive global language distribution database (available through Ethnologue), which is accessible only on a prohibitively expensive licence (Matacic, 2020). At least 10% of languages in the database are given as point estimates rather than polygons. Some languages are classified as 'widespread' or 'immigrant' and recorded outside their area of origin, but not all immigrant populations are included in language range maps. All of these differences between species distribution data and language range maps mean that spatial measures of diversity used in biology cannot always be applied to available language distributions. While species diversity can by evaluated by estimating how many species ranges overlay any particular geographic point, language diversity can only be estimated for grids that are large enough to encompass many nonoverlapping language ranges (Hua et al., 2019).

Due to the complicated relationship between people and languages, range maps may be a poor indication of number of individuals who are fluent in a language. Signed languages provide an illustrative example. There are more than 270 recognised signed languages, ranging from languages used in a single village (e.g., Ban Khor Sign Language in north-eastern Thailand) to nationally recognised languages such as Brazilian Sign Language (also known as Libras, used in education and government, with 200,000 L1 signers). Signed languages often have both L1 signers (e.g., Deaf children who learned to sign as a first language) and fluent L2 signers (e.g., hearing parents of Deaf children who learned to sign as adults). Some local signed languages are used by all community members as part of their communicative repertoire (Nonaka, 2004; Maypilama and Adone, 2013). Many signed languages are endangered (Braithwaite, 2019), but often there is a lack of basic data on distribution and number of signers. Half of the signed languages in Ethnologue have a recorded L1 of zero (Eberhard et al., 2022), but this may reflect lack of information rather than lack of active signers. For example, there is no formal census of Deaf people in Vietnam (Woodward et al., 2015), so the number of L1 signers of Ho Chi Minh City Sign is listed as zero in Ethnologue, although the Endangered Languages Project estimates 45,000 signers (extrapolated from likely incidence of congenital deafness). Half of the signed languages in Glottolog have no recorded level of endangerment, but over three quarters of those with an assessment are considered endangered (Hammarström et al., 2022). Range data are available for very few signed languages, and describing the geographic distribution is

complicated where L1 signers form a small minority within the area of a spoken language. For example, Auslan is distributed across Australia, but is an L1 language for less than 0.04% of people within the Australian population. Many of these complications of interpreting range data will also apply to some spoken languages, for example, for ethnic minority languages distributed within larger populations (e.g., many Romani languages like Domari).

The point to emphasise is not that language distribution data are inadequate in some way, but simply that language range maps differ from species distribution data in several important respects, so care must be taken before applying methods designed for species distribution data to language ranges. One feature of species distribution data that has thus far had little application in analysis of language distribution and diversity is the construction and use of open-access databases of location data submitted by both professionals and amateurs. For example, the Atlas of Living Australia is used to model past, present and future species distributions, using data from professional sources (e.g., herbaria collections and survey data) and 'citizen science' (e.g., species observations submitted by members of the public) (Belbin et al., 2021).

A more fundamental problem with using IUCN-style metrics based on range size and population to identify spatial patterns of languages endangerment is that it risks conflating diversity and endangerment. Languages, like species, show a latitudinal diversity gradient (Mace and Pagel, 1995; Currie et al., 2013). Tropical areas with high year-round productivity have more languages and can sustain languages with smaller L1 speaker population sizes (Hua et al., 2019). In highly productive areas, languages tend to have relatively smaller distributions, so more languages can be 'packed' into a given area. If L1 population size and area are used as indicators of endangerment, then we would expect tropical areas to have both more languages and more endangered languages, making it difficult to disentangle patterns of diversity from patterns of endangerment (Figure 1).

## Global patterns of language endangerment and loss

The IUCN Red List provides a centralised database and a global standard for assessing and communicating species endangerment. Currently, there is no equivalent internationally agreed standard for language endangerment. The standard indicators of species endangerment underlying IUCN ratings – population size and range decline – are problematic for evaluating language endangerment. Instead, language endangerment scales tend to focus on transmission from one generation to the next and domains of use. Several global databases of endangered languages exist, each using a different endangerment scale and having different coverage (Table 1). Ethnologue and Glottolog include all languages, but the UNESCO and the Catalogue of Endangered Languages focus on endangered languages, so languages not included in these databases may be either not endangered (equivalent to 'Least Concern' in IUCN) or currently lacking an assessment (equivalent to 'Data Deficient' in IUCN). While the different scales and databases broadly agree, there are some differences in rankings for individual languages (Figure 2). For example, White Lachi language of Vietnam is listed as 'extinct' in Glottolog but 'endangered' in Ethnologue, and Bahing language of Nepal is listed as 'nearly extinct' in Glottolog but 'Stable' in Ethnologue. The databases also vary in how they recognise and assign endangerment scores to dialects (subdivisions of languages) or macrolanguages (clusters of similar languages or dialects).

IUCN ratings have been used to identify predictors of extinction risk for groups of species, for example, body size and diet breadth (Chichorro et al., 2019). These intrinsic factors interact with external threats, such as habitat reduction, invasive species, disease or climate change, to make some species more vulnerable to decline than others (Purvis et al., 2005; Lee and Jetz, 2011). While identifying correlates of extinction risk does not necessarily translate into conservation action (as conservation efforts tend to be occur in 'crisis management' mode to aimed at rescuing specific species or ecosystems; Cardillo and Meijaard, 2012), it does contribute to a general understanding of patterns of biodiversity loss which may help to inform management strategies or predict future endangerment status (Di Marco et al., 2012).

Is a similar approach to identifying correlates of endangerment valuable for languages? Every endangered language has its own story to tell. Each has a particular history, a unique set of linguistic attributes, specific interactions with other cultures and particular combination of socioeconomic pressures. A quantitative, comparative approach cannot capture all of the nuances of culture, history and environment that have shaped any given language's current state and future trajectory. Instead, the aim of a comparative analysis of endangerment is to detect general trends and shared factors that impact on the fates of many different languages. These shared factors will never give a full account of endangerment for any one language, nor will they have strong predictive power on their own, but they may highlight general risk factors that threaten the vitality of many languages.

Statistical analyses of global databases allow formal testing of hypotheses. For example, it has been suggested that languages having official status in one or more countries will be less likely to be endangered (Thomason, 2015), yet when official status is included in a statistical model, languages with official status are not significantly less endangered than those without official recognition (Bromham et al., 2022). However, it is important to distinguish lack of significant predictive power from lack of influence. For example, there has been a debate over whether major world languages, such as English, act as 'killer' languages (Mühlhäusler, 2002; Skutnabb-Kangas, 2003; Mufwene, 2005), but occurring in a country that has English or any other major world language as an official language is not a significant predictor of endangerment status (Bromham et al., 2022). The lack of a consistent global correlation does not deny that shift to English is a key factor in the decline of some languages (Schaefer and Egbokhare, 1999; Rapatahana and Bunce, 2012): it simply indicates that knowing whether a country has official recognition for a world language does not help us predict the endangerment status of languages.

Identifying correlates of endangerment, whether internal factors or external threats, requires separating out patterns due to shared environment, patterns of relatedness and covariation between traits (Bromham et al., 2018). For example, it has been suggested that polysynthetic languages are particularly vulnerable to decline (Vakhtin and Gruzdeva, 2017). Words in polysynthetic languages are formed from many different morphemes (units of meaning), allowing complex ideas to be coded into single words. Because polysynthetic languages can be challenging for adults to learn, it has been proposed that they are prone to simplification and loss in contact situations (where adults are learning to communicate with speakers of a different language). The challenge in testing this hypothesis is that there are many covarying factors to be untangled.

## (A) Number of endangered languages

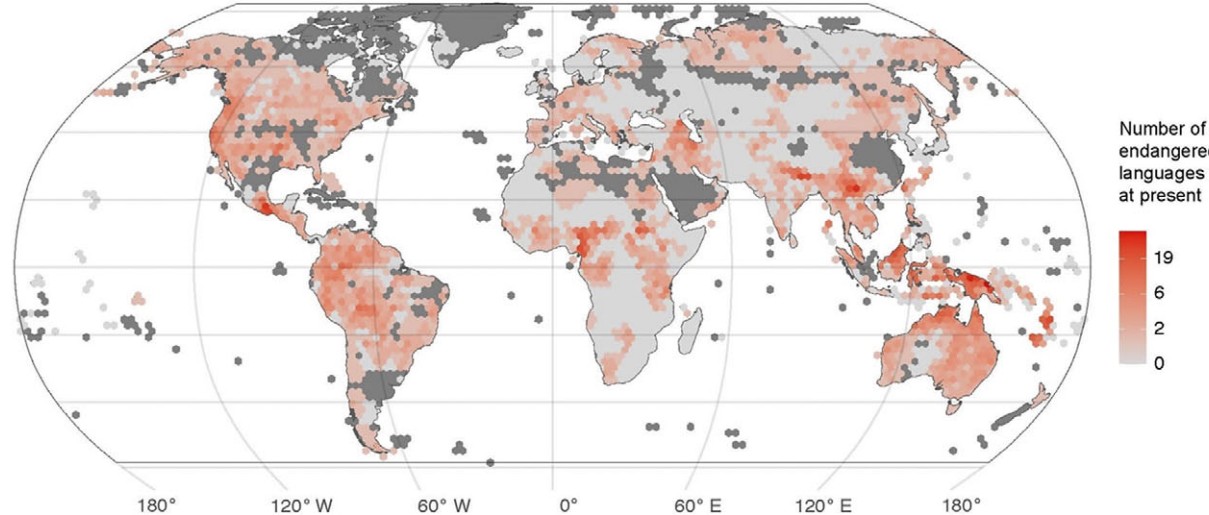

## (B) Proportion of languages that are endangered

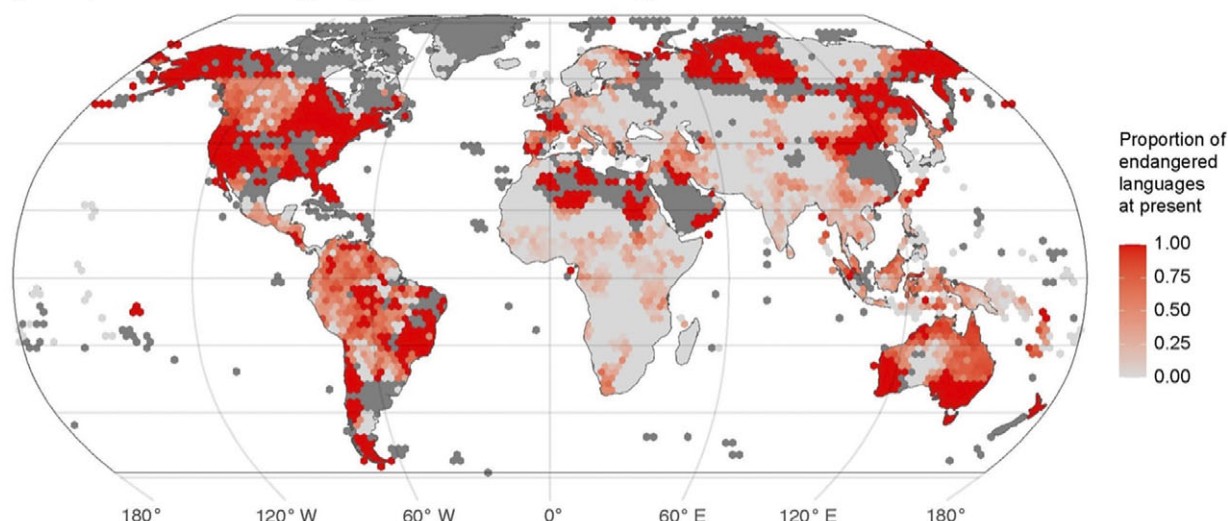

**Figure 1.** Global distribution in the number of endangered languages (A) and the proportion of languages in each grid cell that are endangered (B), for languages with an Expanded Graded Intergenerational Disrupted Scale (EGIDS) rating of 6b to 10, based on a database of 6,511 spoken languages. Figure created by Xia Hua and reproduced from Bromham et al. (2022) under Creative Commons Attribution 4.0 International Licence.

The majority of polysynthetic languages are found in North America, which has suffered one of the highest rates of language loss in the world as a result of brutal colonial suppression of Indigenous languages (Figure 3). The occurrence of so many polysynthetic languages in a region with high rates of language endangerment and loss will cause an association between polysynthesis and endangerment, even if there is no causal connection between complex word structure and language decline.

Furthermore, related languages tend to occur in neighbouring areas which have similar environments and histories, generating a problem of spatial autocorrelation. Including many nearby languages in the same analysis as if they were independent datapoints leads to pseudo-replication, because they essentially sample the same environment and history, inflating the appearance of a correlation between their shared characteristics and their environment. For example, speaker population sizes in North America (where the majority of polysynthetic languages are found) tend to be lower than those in Europe and Asia (where there are few polysynthetic languages), due to environmental and social differences (Lupyan and Dale, 2010; Trudgill, 2017). Since smaller speaker population size is associated with language endangerment, this creates another possible indirect connection between polysynthesis and endangerment. Smaller population size, higher endangerment, a shared colonial history and polysynthesis all broadly co-occur in related families in the same region. If many North American languages were included in a global study of language endangerment, each one would rate relatively high on complexity and level of endangerment, and relatively low on population size, potentially leading to correlations between polysynthesis and endangerment even if endangerment was wholly due to historical factors and not influenced by language complexity.

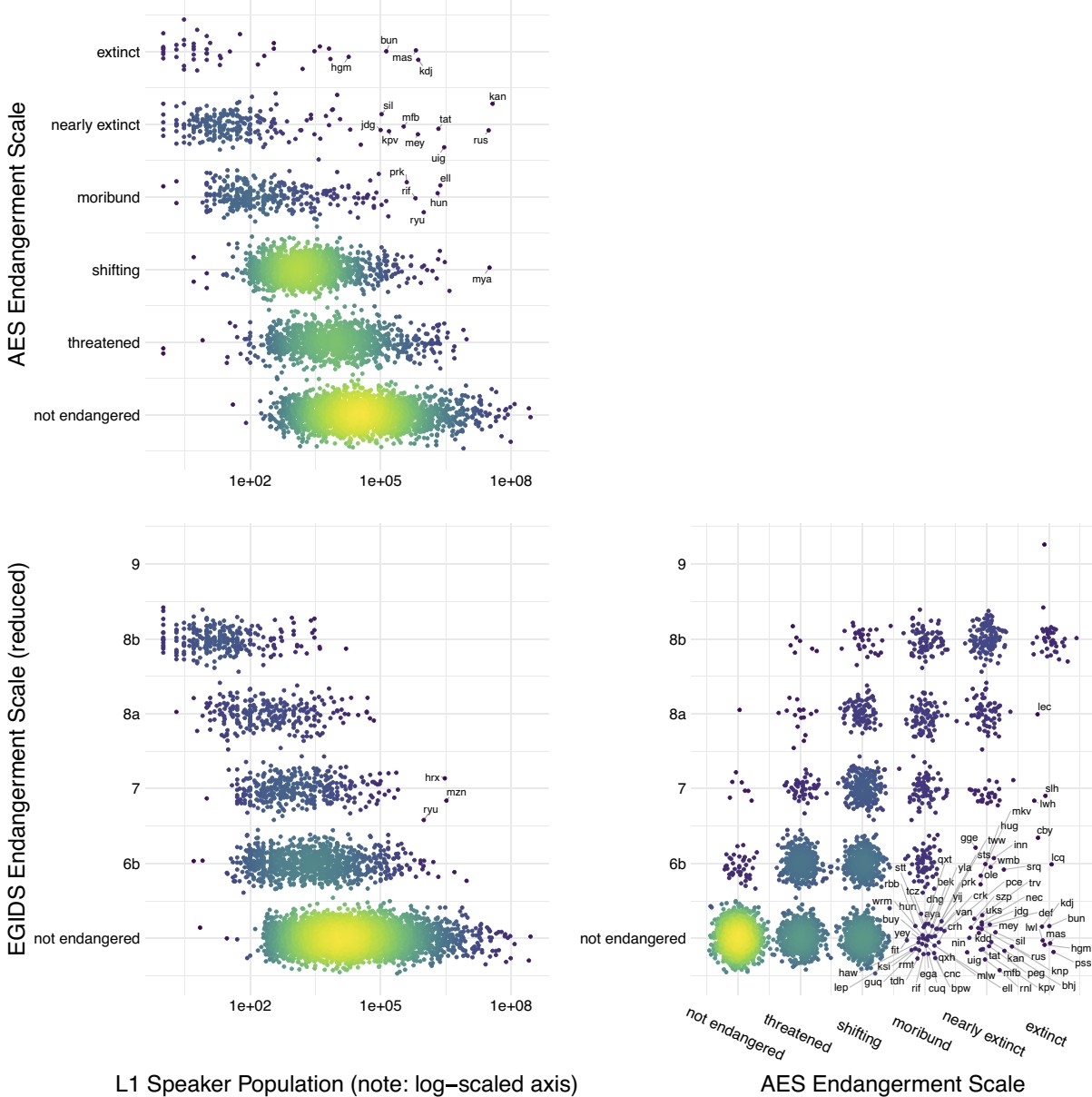

**Figure 2.** Endangerment scales and L1 speaker population size. The relationship between two endangerment scales – the Agglomerated Endangered Scale (AES, used in Glottolog) and the Expanded Graded Intergenerational Disrupted Scale (EGIDS, used in Ethnologue; see Table 1) – and L1 speaker population size. Some languages are identified by three letter ISO-693 codes that act as unique identifiers for languages. These databases are constantly being updated so that this figure represents a snapshot of the data at the time of access (Ethnologue v.17/v.16 and Glottolog v4.2.1). Figure created by Russell Dinnage and reproduced from Bromham et al. (2022) under Creative Commons Attribution 4.0 International Licence.

Fortunately, there are methods designed to untangle such causal interconnections caused by relatedness, proximity and covariation. First, it is essential to take account of the fact that related languages tend to be more similar to each other in many features, even if those features are not structurally related. This problem is known in biology as phylogenetic nonindependence, but is often referred to in cultural evolution studies as Galton's problem (Naroll, 1965). Relationships between languages, gleaned from a taxonomy or phylogeny or any other source of information, can be used to either select datapoints that approximate statistical independence, or to inform a matrix of expected covariance due to shared ancestry (other approaches are discussed in a review of solutions to Galton's problem; see Bromham, 2022).

Second, any test of a hypothesis linking either an intrinsic or extrinsic factor to language endangerment must allow for proximity. Nearby languages will tend to share aspects of their environment, and will often have similar historical influences (e.g., the history of colonisation that impacts all North American languages), potentially generating spurious correlations. For example, in New Guinea, there has been a greater rate of language endangerment in the lowlands than the highlands, potentially due to environmental differences (e.g., mosquito-borne diseases in the lowlands but not the highlands) and history (e.g., patterns of human settlement and land use). Given that lowland areas are likely to have more endangered languages, any environmental factor that differs between the lowlands and highlands could also correlate with language

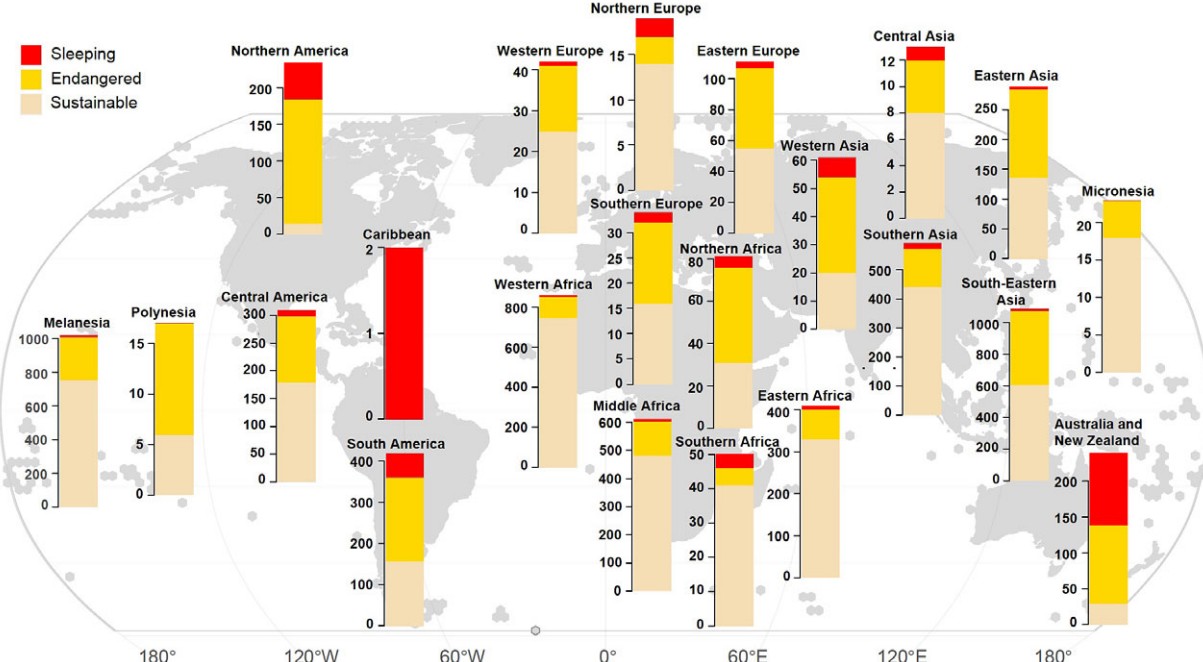

**Figure 3.** Relative number of languages rated as Sustainable (Expanded Graded Intergenerational Disrupted Scale [EGIDS] 1–6a; see Table 1), Endangered (EGIDS 6b–10) or Sleeping (EGIDS 9–10) by region (as defined in the Natural Earth database), based on a database of 6,511 spoken L1 languages. Figure created by Xia Hua and reproduced from Bromham et al. (2022) under Creative Commons Attribution 4.0 International Licence.

endangerment, even if it is not causally connected. There are higher rates of mammal species endangerment in the highlands, causing a negative correlation between species endangerment and language endangerment, when both are measure across grid cells (Turvey and Pettorelli, 2014). But when you take spatial distribution of observations into account, there is no association between species and language endangerment beyond what you would expect from their proximity (Cardillo et al., 2015).

Third, to avoid being led astray by indirect associations between variables, it is important to evaluate the explanatory power of variables above and beyond their covariation with other variables (Roberts and Winters, 2013). For example, languages show a latitudinal diversity gradient: the tropics have more languages, which tend to have smaller populations and more restricted areas (Mace and Pagel, 1995; Sutherland, 2003; Hua et al., 2019). Since population size is correlated with endangerment, we should not be surprised that endangerment correlates with other variables with a latitudinal gradient, such as gross domestic product (GDP) and population density (Kummu and Varis, 2011; Bonds et al., 2012). But when the effect of proximity, relatedness and covariation are taken into account, there is no evidence that lower GDP is associated with higher language endangerment (Sutherland, 2003; Bromham et al., 2022). Comparing the explanatory power of different predictor variables weighs alternative explanations for endangerment. Speaker population size is correlated with population density, which is positively correlated with road density, land-use change and human impact on the environment, and negatively correlated with life expectancy, educational attainment and GDP per capita, but only road density and average years of schooling have a significant, worldwide association with language endangerment, above and beyond their covariation with other variables (Bromham et al., 2022).

An important caveat of broad scale studies of endangerment is that scale matters: trends identified at the global level may not represent the major threats to specific languages or particular areas. The global models explain around a third of the variation in language endangerment, comparable to similar studies of extinction risk in species (Bromham et al., 2022), so the majority of impact is due to other factors, or idiosyncratic aspects of history, or chance. However, identification of global correlates of threat status may identify influences that could be investigated at finer scales. For example, average number of years in school is a significant predictor of language endangerment, over and above other indicators of socioeconomic development and independently of land use or urbanisation. Country-level averages cannot capture regional, ethnic or socioeconomic variation in schooling levels; however, local scale studies have provided similar conclusions. For example, number of years of formal schooling is a predictor of Indigenous language use in a remote Australian community, across all age groups (Bromham et al., 2020). Similar patterns have been reported in North America, where, like Australia, high-stakes testing have shifted focus to English competency at the expense of Indigenous language proficiency (e.g., McCarty, 2003; Wyman et al., 2010; Combs and Nicholas, 2012).

## Predicting future patterns of languages loss

Identification of correlates of endangerment can provide a way of predicting future patterns of language endangerment and loss. For example, correlates of species extinction risk can be used to identify species with high 'latent risk', having factors that predispose them to endangerment even if they are not currently threatened with extinction (Cardillo et al., 2006). Can a similar approach be used for languages? Environmental correlates of language endangerment, such as climate or land-use change, can be used to project future pressures on languages (Bromham

et al., 2022). However, the impact of such factors on language is small compared with demographic shift. If a language is not being learned by children, it will cease to have L1 speakers once the current generation have died. Language endangerment has been modelled using patterns of decline in speaker numbers (Amano et al., 2014), but these data are available for relatively few of the world's languages. While population trajectories for species can be modelled using information on species characteristics such as reproductive rate and age-specific mortality, the change in the number of speakers over time is dependent on a complex web of social factors which may change rapidly over time, so may be hard to extrapolate even given estimates of language transmission rates in previous generations. Knowing the population size of the Dom people (counted in millions, spread across the Middle East, north Africa and Europe), or the birth and death rate of those populations, would not help you predict the endangerment status of the Domari language ('Moribund' in Glottolog and 'Endangered' in Ethnologue). Nonetheless, while we may lack the ability to predict the fate of any given language based only on known correlates of language endangerment, we may be able to identify 'hotspots' of future language loss by identifying areas where factors known to influence language endangerment are operating, such as increase in road density or implementation of high-stakes educational testing in economically dominant languages.

Forms of future projection commonly employed in conservation biology appear to be of limited use in language endangerment and loss. Population viability analysis is used to project species' trajectories, by modelling birth and death rates, in order to evaluate species endangerment and inform management strategies (Akçakaya and Sjögren-Gulve, 2000; Brook et al., 2000). This approach works best when there are accurate data on reproductive parameters and population growth rates, and where these parameters can be reliably extrapolated to future prediction (Coulson et al., 2001). However, this approach will not work for languages because vitality of languages is based on the transmission of language rather than human reproduction, so language speaker trends may be unconnected to population demography. For example, the number of L1 speakers of the Australian language Bardi declined for over a century even as the Bardi population was growing, and the Bardi language is now critically endangered (Bowern, 2012).

Projecting future range shifts under climate change and land-use modification is an important part of species conservation planning, and a suite of analytical tools have been developed to model the future availability of suitable habitat. For example, many species are limited in extent by their range of thermal tolerance, so projecting temperature change under future climate scenarios informs likely distribution of suitable habitat in future (Schwartz, 2012; Bonebrake et al., 2018). However, such approaches are unlikely to be useful for languages. Environmental features such as mean growing season do shape broad scale patterns of language diversity (Nettle, 1998), and climate change may impact Indigenous and minority language populations in ways that may threaten language vitality, for example, through population movement or sea level change (Dunn, 2017; Addaney et al., 2022; Brown and Middleton, 2022). However, compared with socioeconomic factors and educational policy, the direct influence of climate on 'language niche' is unlikely to be a major determinant of the future vitality of most endangered languages (Antunes et al., 2020).

There is a growing move to use macroecological analysis of patterns of endangered species to prioritise conservation efforts,

in order to optimise protection for areas that contain a greater proportion of biodiversity, or would help to conserve a greater number of threatened species (Margules and Pressey, 2000; Allan et al., 2019). Similarly, identification of language hotspots has been used to focus attention on areas where the greatest diversity of language is at risk of loss, potentially focusing effort on developing programmes that will encourage documentation and support language vitality (Anderson, 2011). Clusters of endangered languages suggest area-wide influences on vitality (Lee et al., 2022), although in identifying meaningful patterns it is essential to control not only for language diversity of an area but also population size and range size, which show geographic patterns (Hua et al., 2019). However, unlike species conservation that is entirely dependent on 'top-down' management (imposing strategies aimed at safeguarding declining populations), language vitality also depends critically on 'bottom-up' support from within communities to encourage use of their heritage language. For example, 'language nests' – where community elders contribute to language immersion programmes in schools or childcare – illustrate the combination of support of local, regional or national governments with community-based involvement (Hinton et al., 2018). Language revitalisation strategies may combine both top-down and bottom-up strategies, for example, if academics work with community members to develop online tools and government invests in overcoming the 'digital divide' that sees many indigenous communities with less access to online resources (Cunliffe and Herring, 2005). Identification of hotspots should not be used to preference investment in some languages over others. Instead, they can be used to highlight that language diversity needs to be valued by authorities at local, regional and national levels, in order to empower and support individuals and communities to safeguard and reinvigorate their linguistic heritage.

## Conclusion

Language diversity and biodiversity show some striking similarities, including, unfortunately, their current catastrophic rates of loss. There is room for biologists and linguists to join forces to share useful tools and insights in understanding and protecting endangered species and languages. However, tools from conservation biology should not be adopted uncritically into studies of language diversity without examining the degree to which the method captures patterns and processes relevant to language endangerment and loss. Like studies of species endangerment, broad-scale analyses of language endangerment patterns do not provide a comprehensive picture of the current vitality and future prospects of any specific language. However, they contribute to the understanding of patterns of language diversity and loss, and serve to highlight important interventions to maintain global language diversity, such as making sure that education programmes support, not erode, language diversity.

**Open peer review.** To view the open peer review materials for this article, please visit http://doi.org/10.1017/ext.2022.3.

**Acknowledgements.** I would like to thank Rhiannon Schembri, Xia Hua, Hedvig Skirgård, Felicity Meakins, Sam Passmore, Mark Vellend and Marcel Cardillo for taking the time to read and comment on the manuscript.

**Financial support.** This research received no specific grant from any funding agency, commercial or not-for-profit sectors.

**Competing interest.** The author declares none.

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
