## [Reviewer Report]

*Comments to Author*: This is a very clear and interesting overview of the current state of affairs for language endangerment, viewed through a biological / ecological lens. The author discusses similarities and differences between languages and biological species, state of research, and what biological methods do (and don't do).

Two really big differences are mentioned, and these are a welcome introduction to the discourse on biological approaches to language endangerment: one is the role that *people* have, and how agentive speakers and signers of endangered languages are. Given that, it would be great to bring in some of that material earlier, e.g. in the introductory overview on p3. Another is the discussion of signed languages, also commonly left out of discourse on endangerment (I was looking for a reference in the first paragraph of p4, but it came later - would it be possible to foreground this point more?) Is it possible to cite more Indigenous perspectives on endangerment? There is a growing body of academic work on this topic, as well as work for general audiences; cf. MK Turner's Iltyem Iltyem, for example. Work by Wes Leonard, Teresa McCarty, Jenny Davis, and others would also be good to cite in this overview, since their perspectives are important for those working on language "at a distance".

p4, last para - Etruscan vs Latin; ie Etruscan left no descendants, whereas Latin did.

p7, Range Size: cf. Bowern and Dockum (forthcoming + 2022 LSA presentation) on issues of language map reading and interpretation. p8 I expect that similar points may apply to species ranges too, but since humans typically live in clusters (settlements, towns, villages, cities, etc), putting a language on a map to depict a continuous range is even more an abstraction.

p12 - middle para maybe reference the CHIELD (Causal Hypotheses Database): https://academic.oup.com/jole/article/5/2/101/5821004

p15, another relevant point, if there's space, is the digital divide, especially the role this plays in countries where Indigenous populations are prevented from using their languages easily online. e.g. the need for predictive text, language support on social media (and the consequences for privacy, etc).

---

## [Reviewer Report]

*Comments to Author*: The author presents a review of how methods developed in biology to categorize how species are endangered has been applied to language diversity, considering both similarities and differences of these fields. The topic is very interesting and timely, given the high percentage of endangered languages. The text also reads very fluently and pleasant. Therefore, I would highly recommend this manuscript for publication given my comments are adequately addressed. Most of my comments are fairly minor and require just some improvements in the text, so I expect that the revision will be overall feasible.

Major points:

- Page 7-8: I understand the difficulties in drawing range maps might not be as easily achieved as the author describes. However, species range maps are also plagued with similar problems, such as invasive species and local extirpations due to human activities. Some attempts at covering this have caused the development of invasive species databanks (e.g. https://glonaf.org/), sometimes based on citizen science (e.g. GBIF https://www.gbif.org/). Hence, maybe the development of similar data banks for immigrant, L2, non-taught languages might be the way to go. Is there any citizen science initiative for languages?

- Based on the various data bases and classification schemes (table 1), it might be worth discussing how a potential unification of classification could be beneficial to language diversity assessments. This is considering that at least for species, the widespread use of IUCN criteria is very important. There are some national or provincial red listings, which often follow IUCN criteria but ignore occurrences on other countries or regions. This can be confusing, but such listings are used only locally. It would be interesting to know how these classifications of languages are used and whether there are initiatives to unify them?

- Pages 10-11: It is important to bring caution on interpreting correlations and controlling for confounding effects, autocorrelation and contingency, which are problems not particular to languages and should be checked in every scientific study. With that said, it seems that it would be ungently needed to develop mechanistic or agent-based models for language survival to actually disentangle correlation from causation. Could the author discuss any attempt at developing agent-based models for language diversity and survival? In conservation, there is an increasing demand for agent-based and mechanistic models for biodiversity to actually deal with the lack of causality in correlations. This necessity is strongly highlighted in the IPBES (Intergovernmental Science-Policy Platform on Biodiversity and Ecosystem Services) report, for example. Is there any similar movement in linguistics (to become more mechanistic)? A quick search yielded a few promising papers (e.g. Castelló et al. 2013; Civico 2019).

- Page 14: so if PVA does not entirely work for languages, how would a Language Viability Analysis (LVA) could look like? How one could model the number of L1 speakers based on socioeconomic and educational besides demographic parameters?

- Pages 14-15: the niche discussion: I understand that environmental variables might not be the best ones to predict language occurrence or threat, which is similar for some species. However, if socioeconomic factors and educational policies are really the key factors, one could include such variables as ‘niche’ variables in similar models. What I mean is that the methodology of niche modelling can use variables other than environmental ones. Maybe worth discussing which exactly such variables would be for language. Was this already attempted? Please, clarify.

Minor points:

- Abstract: in the first line, I would say is not a nice tone to relate which is more in peril, language or species diversity. A key difference is that we do not know yet how many species there are and most of these by far and wide are in the tropics, which are being massively deforested. This issue of unknown diversity is thematized early in page 4. Hence, it could be that actually the species diversity is more endangered. But this is besides the point of the paper. So I suggest the authors to refrain from comparative language.

- Please use line numbers

- Last sentence of second paragraph of page 6: ‘which is’ duplicated, please delete one.

- Figure 2: Please, use lettering for the different panels and explain each panel in the caption. Increase font size in the panels. Also, why is there a language in level 9 in the bottom right panel and not in the bottom left panel?

- First paragraph at page 15: there is a comma space period in the end of the paragraph. Is there something missing from the sentence?

References:

Castelló, X., Loureiro-Porto, L. and San Miguel, M., 2013. Agent-based models of language competition. International journal of the sociology of language, 2013(221), pp.21-51.

Civico, M., 2019. The dynamics of language minorities: Evidence from an agent-based model of language contact. Journal of Artificial Societies and Social Simulation, 22(4), p.27.

---

## [Editor Report]

*Comments to Author*: I agree with the reviewers that the topic is important, timely, and highly relevant, and that the manuscript is very well developed, but I also agree that there is room for improvement, especially regarding the issue of language mapping and threat classification. While the author provides quite thorough and sound assessment of challenges that such efforts face, it would be useful for the readership and the field to indicate some ways forward related to range maps and threat classifications, in line with the comments below.

Reviewer 1:

This is a very clear and interesting overview of the current state of affairs for language endangerment, viewed through a biological / ecological lens. The author discusses similarities and differences between languages and biological species, state of research, and what biological methods do (and don't do).

Two really big differences are mentioned, and these are a welcome introduction to the discourse on biological approaches to language endangerment: one is the role that *people* have, and how agentive speakers and signers of endangered languages are. Given that, it would be great to bring in some of that material earlier, e.g. in the introductory overview on p3. Another is the discussion of signed languages, also commonly left out of discourse on endangerment (I was looking for a reference in the first paragraph of p4, but it came later - would it be possible to foreground this point more?) Is it possible to cite more Indigenous perspectives on endangerment? There is a growing body of academic work on this topic, as well as work for general audiences; cf. MK Turner's Iltyem Iltyem, for example. Work by Wes Leonard, Teresa McCarty, Jenny Davis, and others would also be good to cite in this overview, since their perspectives are important for those working on language "at a distance".

p4, last para - Etruscan vs Latin; ie Etruscan left no descendants, whereas Latin did.

p7, Range Size: cf. Bowern and Dockum (forthcoming + 2022 LSA presentation) on issues of language map reading and interpretation. 

p8 I expect that similar points may apply to species ranges too, but since humans typically live in clusters (settlements, towns, villages, cities, etc), putting a language on a map to depict a continuous range is even more an abstraction.

p12 - middle para maybe reference the CHIELD (Causal Hypotheses Database): https://academic.oup.com/jole/article/5/2/101/5821004

p15, another relevant point, if there's space, is the digital divide, especially the role this plays in countries where Indigenous populations are prevented from using their languages easily online. e.g. the need for predictive text, language support on social media (and the consequences for privacy, etc).

Reviewer 2:

The author presents a review of how methods developed in biology to categorize how species are endangered has been applied to language diversity, considering both similarities and differences of these fields. The topic is very interesting and timely, given the high percentage of endangered languages. The text also reads very fluently and pleasant. Therefore, I would highly recommend this manuscript for publication given my comments are adequately addressed. Most of my comments are fairly minor and require just some improvements in the text, so I expect that the revision will be overall feasible. 

Major points:

- Page 7-8: I understand the difficulties in drawing range maps might not be as easily achieved as the author describes. However, species range maps are also plagued with similar problems, such as invasive species and local extirpations due to human activities. Some attempts at covering this have caused the development of invasive species databanks (e.g. https://glonaf.org/), sometimes based on citizen science (e.g. GBIF https://www.gbif.org/). Hence, maybe the development of similar data banks for immigrant, L2, non-taught languages might be the way to go. Is there any citizen science initiative for languages? 

- Based on the various data bases and classification schemes (table 1), it might be worth discussing how a potential unification of classification could be beneficial to language diversity assessments. This is considering that at least for species, the widespread use of IUCN criteria is very important. There are some national or provincial red listings, which often follow IUCN criteria but ignore occurrences on other countries or regions. This can be confusing, but such listings are used only locally. It would be interesting to know how these classifications of languages are used and whether there are initiatives to unify them?

- Pages 10-11: It is important to bring caution on interpreting correlations and controlling for confounding effects, autocorrelation and contingency, which are problems not particular to languages and should be checked in every scientific study. With that said, it seems that it would be ungently needed to develop mechanistic or agent-based models for language survival to actually disentangle correlation from causation. Could the author discuss any attempt at developing agent-based models for language diversity and survival? In conservation, there is an increasing demand for agent-based and mechanistic models for biodiversity to actually deal with the lack of causality in correlations. This necessity is strongly highlighted in the IPBES (Intergovernmental Science-Policy Platform on Biodiversity and Ecosystem Services) report, for example. Is there any similar movement in linguistics (to become more mechanistic)? A quick search yielded a few promising papers (e.g. Castelló et al. 2013; Civico 2019). 

- Page 14: so if PVA does not entirely work for languages, how would a Language Viability Analysis (LVA) look like? How one could model the number of L1 speakers based on socioeconomic and educational besides demographic parameters?

- Pages 14-15: the niche discussion: I understand that environmental variables might not be the best ones to predict language occurrence or threat, which is similar for some species. However, if socioeconomic factors and educational policies are really the key factors, one could include such variables as ‘niche’ variables in similar models. What I mean is that the methodology of niche modelling can use variables other than environmental ones. Maybe worth discussing which exactly such variables would be for language. Was this already attempted? Please, clarify.

Minor points:

- Abstract: in the first line, I would say is not a nice tone to relate which is more in peril, language or species diversity. A key difference is that we do not know yet how many species there are and most of these by far and wide are in the tropics, which are being massively deforested. This issue of unknown diversity is thematized early in page 4. Hence, it could be that actually the species diversity is more endangered. But this is besides the point of the paper. So I suggest the authors to refrain from comparative language.

- Please use line numbers

- Last sentence of second paragraph of page 6: ‘which is’ duplicated, please delete one. 

- Figure 2: Please, use lettering for the different panels and explain each panel in the caption. Increase font size in the panels. Also, why is there a language in level 9 in the bottom right panel and not in the bottom left panel?

- First paragraph at page 15: there is a comma space period in the end of the paragraph. Is there something missing from the sentence?

References:

Castelló, X., Loureiro-Porto, L. and San Miguel, M., 2013. Agent-based models of language competition. International journal of the sociology of language, 2013(221), pp.21-51.

Civico, M., 2019. The dynamics of language minorities: Evidence from an agent-based model of language contact. Journal of Artificial Societies and Social Simulation, 22(4), p.27.

---

## [Reviewer Report]

*Comments to Author*: Thank you for the attention to the previous comments; everything has been sufficiently addressed

---

## [Editor Report]

*Comments to Author*: Thank you for submitting the revised version of your manuscript. Based on the evaluation of the changes made in the manuscript and comments of reviewers, I think that the manuscript can be accepted in its present form.

There are several minor typos remaining in the text, but this can be corrected at the typesetting and proof stage (e.g., double opening parentheses in L44, L52, L190 and L218, missing space in L56, two sets of parentheses with citations in L200).